# Formulation and Characterization of Chitosan-Decorated Multiple Nanoemulsion for Topical Delivery In Vitro and Ex Vivo

**DOI:** 10.3390/molecules27103183

**Published:** 2022-05-17

**Authors:** Muhammad Rehan Malik, Fatemah Farraj Al-Harbi, Asif Nawaz, Adnan Amin, Arshad Farid, Mohammed Al Mohaini, Abdulkhaliq J. Alsalman, Maitham A. Al Hawaj, Yousef N. Alhashem

**Affiliations:** 1Gomal Centre of Pharmaceutical Sciences, Faculty of Pharmacy, Gomal University, Dera Ismail Khan 29050, Pakistan; malicrehan@gmail.com (M.R.M.); adnan.amin@gu.edu.pk (A.A.); 2College of Science, Princess Nourah Bint Abdulrahman University, Riyadh 84428, Saudi Arabia; ffalharbi@pnu.edu.sa; 3Gomal Center of Biochemistry and Biotechnology, Gomal University, Dera Ismail Khan 29050, Pakistan; 4Basic Sciences Department, College of Applied Medical Sciences, King Saud bin Abdulaziz University for Health Sciences, Ahsa 31982, Saudi Arabia; mohainim@ksau-hs.edu.sa; 5King Abdullah International Medical Research Center, Ahsa 31982, Saudi Arabia; 6Department of Clinical Pharmacy, Faculty of Pharmacy, Northern Border University, Rafha 91911, Saudi Arabia; kaliqs@gmail.com; 7Department of Pharmacy Practice, College of Clinical Pharmacy, King Faisal University, Ahsa 31982, Saudi Arabia; hawaj@kfu.edu.sa; 8Clinical Laboratory Sciences Department, Mohammed Al-Mana College for Medical Sciences, Dammam 34222, Saudi Arabia; yousefa@machs.edu.sa

**Keywords:** chitosan, multiple nanoemulsion, topical delivery, itraconazole

## Abstract

In the present study, chitosan-decorated multiple nanoemulsion (MNE) was formulated using a two-step emulsification process. The formulated multiple nanoemuslion was evaluated physiochemically for its size and zeta potential, surface morphology, creaming and cracking, viscosity and pH. A Franz diffusion cell apparatus was used to carry out in vitro drug-release and permeation studies. The formulated nanoemulsion showed uniform droplet size and zeta potential. The pH and viscosity of the formulated emulsion were in the range of and suitable for topical delivery. The drug contents of the simple nanoemulsion (SNE), the chitosan-decorated nanoemulsion (CNE) and the MNE were 71 ± 2%, 82 ± 2% and 90 ± 2%, respectively. The formulated MNE showed controlled release of itraconazole as compared with that of the SNE and CNE. This was attributed to the chitosan decoration as well as to formulating multiple emulsions. The significant permeation and skin drug retention profile of the MNE were attributed to using the surfactants tween 80 and span 20 and the co-surfactant PEG 400. ATR-FTIR analysis confirmed that the MNE mainly affects the lipids and proteins of the skin, particularly the *stratum corneum*, which results in significantly higher permeation and retention of the drug. It was concluded that the proposed MNE formulation delivers drug to the target site of the skin and can be therapeutically used for various cutaneous fungal infections.

## 1. Introduction

Currently, a great interest has been revived in the development of new or innovative drug carrier systems for existing drug molecules. These developments not only increase efficacy and safety efficiency of the medication but also improve therapeutic benefit and patient compliance as well [1]. The main intention of a topical drug delivery system is to provide the drug into the skin layers with minimum systemic circulation, at a predetermined rate, and through the skin, with minimum inter- and intrapatient variability [2]. Presently, topical drug delivery is very important and an acceptable drug application method [1].

Topical drug delivery is very important for the administration of various drugs because it enhances the therapeutic efficacy of the drugs as well as reducing their systemic side effects [3]. Other than the oral route, drug administration by the transdermal route is a desirable and more preferable one. Topical drug administration counters the drawbacks related to oral drug administration such as gastrointestinal degradation, hepatic clearance and hepatic first-past metabolism [4]. However, topical drug delivery offers poor permeability, and therefore reduced bioavailability, due to the stratum corneum, which acts as a skin barrier [5,6].

Skin considers all drugs and excipients as external and prevents their entrance inside the body because the skin is the body’s primary defensive layer. Different techniques have been investigated to counter this barrier-like nature of the skin. Therefore, various novel techniques have been developed to increase the penetration of drugs through the skin and to improve percutaneous absorption. These techniques include vehicle systems, novel drug delivery systems, penetration enhancers, etc. [7].

Microemulsions and nanoemulsions, as topical formulations, are considered potential approaches for delivering poorly water-soluble drugs, such as itraconazole (ITZ), to the epidermis and the deeper skin tissues. Actually, these formulations provide solubilized ITZ, which allows the drug to be absorbed quickly into the skin tissues through the stratum corneum’s lipophilic domain [8]. Chitosan, being deacetylated chitin, has many industrial applications, including applications in agriculture, food, cosmetics and pharmaceuticals, with diverse and characteristic physiological and biological properties. These applications are possible due to chitosan’s positive charge [9,10]. The cationic feature of chitosan was considered in designing systems for the delivery of particulate drugs. Chitosan has been successfully used in a number of biomedical applications, including as an artificial skin substitute because it is generally safe and has no harmful effects after implantation in tissue [11]. The cationic feature of chitosan allows it to interact with polyanions, a process that has been used to create drug carrier systems via complexation resulting in nano- and microparticles [12]. It has also been studied as a superior mucoadhesive cationic polymer because of its ability to establish molecular interactions with anionic mucin glycoprotein (depending on environmental pH), which is determined by the formation of hydrogen bonds or ionic interactions with anionic mucin glycoprotein [13].

ITZ is a triazole-based antifungal drug commonly used to treat cutaneous mycoses because of its high activity against a wide range of pathogenic fungi, including dermatophytes and yeast that cause disease [14]. ITZ is usually given orally at a dosage of 200–400 mg/day to treat fungal infections in clinical practice [15,16]. However, oral administration of itraconazole can frequently cause hepatocellular and cholestatic damage from systemic exposure [17,18]. Apart from avoiding ITZ’s side effects, there is a dire need for the drug to be administered topically into the stratum corneum, which is the skin’s outermost layer and the primary focus of treatments for cutaneous mycoses [19]. In spite of this, due to the very low aqueous solubility of ITZ of 1 ng/mL, its topical dosage forms are scarcely available on the market [20]. Liposomes, nanostructured lipid carriers and solid-lipid nanoparticles are some of the topical dosage forms that have been studied for poorly soluble drugs [21,22]. These are particulate or colloidal carrier systems that help to solubilize poorly soluble drugs to some extent but have drawbacks in terms of delivering the drugs to the stratum corneum and deeper skin tissues [23].

One of the best approaches is to prepare a nanoemulsion for topical drug delivery. From this perspective, nanoemulsions have attracted considerable curiosity as new dosage forms because of their substantial solubilizing ability both for hydrophilic and lipophilic compounds [18,24]. Due to their smaller size and flexible nature, nanoemulsions can easily penetrate through the skin. The main drawback with the nanoemulsion is leaching and faster release of the drug. To overcome this problem, multiple nanoemulsions were prepared containing itraconazole (a poorly water-soluble drug) and chitosan as the controlling polymer.

The present study was to prepare chitosan-decorated multiple nanoemulsions, in order to control and regulate the premature release of the drug at the skin’s surface.

## 2. Materials and Methods

### 2.1. Materials

Itraconazole (ITZ) was used as the drug of interest. Chitosan (Sigma, Darmstadt, Germany) was used as rate-controlling polymer. Olive oil (Virgina, Barcelona, Spain) was used as oil phase. Span 20 and Tween 80 (Sigma, Germany) were used as surfactants and PEG 400 as a co-surfactant (Sigma-Aldrich, Darmstadt, Germany). All the chemicals used were of analytical grade.

### 2.2. Methodology

Preparation of Multiple Nanoemulsions.

O/W/O multiple nanoemulsions were prepared by means of a two-step emulsification process.

#### 2.2.1. Primary O/W Nanoemulsion

The oil phase consists of olive oil (29.5% *w*/*w*), span 20 (1% *w*/*w*) and ITZ (1% *w*/*w*) initially mixed for 15 min using a magnetic stirrer. After complete dissolving of the drug in the oil phase, the oil phase was added to the aqueous phase, consisting of distilled water 78.5% *w*/*w*, chitosan (2% *w*/*w*), PEG 400 (4% *w*/*w*) and Tween 80 (1% *w*/*w*). After this, the mixture was then homogenized using a high-share mixer (Ultra-Turrax T 24 Basic, IKA, Staufen im Breisgau, Germany) at 10,000 rpm for 7 min.

#### 2.2.2. Final O/W/O Multiple Nanoemulsion

The primary O/W nanoemulsion (already prepared) was added directly to the oil phase consisting of olive oil (44% *w*/*w*) and span 20 (1% *w*/*w*), with a weight ratio of 1:2 (nanoemulsion/oil) (Figure 1 and Table 1). After this, the mixture was then homogenized using a high-share mixer at 10,000 rpm for 7 min.

### 2.3. Characterization of the Multiple Nanoemulsions

#### 2.3.1. Globule Size and Zeta Potential

Samples of the multiple nanoemulsion were placed in disposable sizing cuvettes. After this, dynamic light scattering (Malvern zetasizer, Malvern, UK) was used to evaluate the size of droplets as well as the zeta potential of SNE, CNE and MNE [25].

#### 2.3.2. Surface Morphology

Surface morphology of the prepared formulation was examined using a simple light microscope (CX41RF, OLYMPUS, Tokyo, Japan) equipped with a photographic camera (5 megapixel). Under the microscope, a drop of the prepared formulation was placed on the slide and observed. After that, their photographs were taken.

#### 2.3.3. Drug Content Determination

The drug content of multiple nanoemulsions was determined by dissolving 2 g of formulation in 10 mL phosphate buffer of pH 5.5. The sample was then centrifuged at 14,000 rpm for 30 min. After this, the clear liquid was taken and assayed spectrophotometrically at λ max 262 nm. The absorbance obtained was then converted to its corresponding concentration using a calibration curve, and then the exact amount of the drug in the formulation was calculated.

#### 2.3.4. Entrapment Efficiency (%) Determination

The concentration of unentrapped drug (free drug) in the formulation was measured to establish the percentage drug encapsulation efficiency. This is important because it affects the drug molecule’s release properties. After separating the entrapped drug from the nanoemulsion formulation, the following Equation (1) was used to calculate the amount of drug encapsulated per unit weight of formulation:%EE = (amount of drug added − free (unentrapped) drug)/(amount of drug added) × 100(1)

#### 2.3.5. Creaming and Cracking

A 30 mL sample of each multiple nanoemulsion (MNE) was taken and was put in a glass bottle having a screw cap (height 65 mm and inner diameter 25 mm), left standing for 1 day at 25 ± 2 °C and then examined for physical appearances. Cracking is a physical instability and can be referred to as the permanent/irreversible partition or separation of the internal/dispersed phase (where oil and water are clearly separated) at the top of the emulsion. If the emulsions are separated into cream and serum layers, the percentage of creaming was determined by calculating the cream layer height (top layer) and the total emulsion height using the given Equation (2).
Creaming (%) = 100 × Height of cream layer/Total height of emulsion(2)

#### 2.3.6. pH of the Multiple Nanoemulsions

The pH of the prepared formulations was determined using digital pH meter (Milwaukee, Mi 150) at room temperature. The same procedure was repeated three times and the result for the formulation was the average value of three readings [25].

#### 2.3.7. Viscosity Measurement

The viscosity of the optimized formulation was determined via Brookfield viscometer (DV-1+ viscometer) (LVDV-1+, LR 99102) without any dilution. Spindle 1 and 2 were used to determine the viscosity of prepared formulations at 30 °C. The spindle was rotated for 1 min at 30 rpm and the reading was noted. The same procedure was conducted in triplicate [25].

### 2.4. In Vitro Release

A Franz-type diffusion cell apparatus was used to carry out in vitro drug release study of the optimized formulations. The cell consists of two compartments, the donor and the recipient. In this study 1 g of drug-loaded formulation was placed in the donor compartment. Cellulose acetate filter was placed between donor and recipient compartment of a Franz-type diffusion cell apparatus having a diffusion area of 0.77 cm^2^ and a diffusion cell volume of 5 mL. The donor compartment was initially empty, whereas the recipient compartment was filled with phosphate buffer pH 5.5 (in simulation of skin pH), which served as the receptor medium or dissolution medium. The temperature of the apparatus was maintained at 32 °C. The receiver compartment was continuously stirred via magnetic stirrer at a speed of 100 rpm. After this, the prepared formulation was spread over the cellulose acetate filter in the donor compartment. Then, 1 mL of sample was withdrawn intermittently for 24 h and replaced with same volume of fresh dissolution media. These samples were than analyzed spectrophotometrically at 262 nm. The amount of drug diffused at specific time intervals was determined and plotted against time. The same procedure was conducted in triplicate and cumulative % drug release was calculated [26].

### 2.5. In Vitro Permeation

In vitro permeation was evaluated using rabbit skin, after ethical approval from the research committee of Gomal University D.I.Khan.

### 2.6. Preparation of Rabbit Skin

Rabbits were slaughtered and hair of abdominal region was removed by using hair removing lotion. Surgical seizure was used to completely remove the skin from the abdominal region. Subcutaneous fats were first surgically removed and washed with normal saline (0.9% NaCl) solution. The skin was then wrapped in aluminum foil and preserved at −20 °C until it was needed again.

For the permeation study, a Franz-type diffusion cell apparatus was used. In this experiment, 1 g of drug-loaded nanoemulsion/multiple nanoemulsion was placed in the donor compartment. Rabbit skin without hair was placed between donor and recipient chamber of Franz-type diffusion cell apparatus. The skin was placed in such a way that its epidermis faced the donor chamber, while the dermal side faced the recipient compartment. The donor compartment was initially empty, whereas the recipient compartment was filled with phosphate buffer pH 7.4 (in simulation of blood pH), which was used as a receptor medium. Temperature of the apparatus was maintained at 37 °C. The receiver compartment was continuously stirred via magnetic stirrer at a speed of 100 rpm. After this, the prepared nanoemulsion formulation was spread over rabbit skin in the donor compartment. Then, 1 mL of sample was withdrawn intermittently for 24 h and replaced with same volume of fresh dissolution media. These samples were then analyzed spectrophotometrically at 262 nm. The amount of drug diffused at specific time intervals was determined and plotted against time. The same procedure was conducted in triplicate and cumulative % drug permeation was calculated [27].

### 2.7. Skin Retention

The amount of drug retained in the skin was calculated at the end of the permeation test. The skin was removed from the cell and cleaned with 0.9% normal saline solution. After this, it was lightly wiped with surgical gauze to remove remaining formulations. The permeation area was then excised and measured. The retained drug content was extracted by cutting the skin into small pieces and vortexing for 3 h in a suitable solvent (5 mL methanol + 15 mL phosphate buffer pH 5.5). After vortexing, the extraction was centrifuged at 14,000 rpm for 5 min. The clear liquid was then analyzed using UV/visible spectrophotometer at λ max 262 nm via phosphate buffer of pH 5.5 as a blank.

### 2.8. ATR-FTIR Analysis of Skin

The ATR-FTIR spectra of untreated skin and skin treated with simple nanoemulsion (SNE), chitosan-decorated nanoemulsion (CNE) and the MNE formulation were taken using an IR spectrophotometer (Perkin Elmer, UK). The samples were placed on zinc selenide crystal and then scanned between 4000 and 400 cm^−1^ [28].

### 2.9. Statistical Analysis

All experiments were conducted in triplicate and the results reported as mean average ± SD. The student’s *t* test and one-way ANNOVA were used for the statistical analysis. The test was done using GraphPad prism 10 software. If the *p* value was less than 0.05, the findings were considered statistically significant [29].

## 3. Results and Discussion

### 3.1. Characterization of the Multiple Nanoemulsions

The prepared nanoemulsions and multiple nanoemulsions were characterized for size and zeta potential, surface morphology, drug content, percent encapsulation efficiency, in vitro release and permeation studies.

#### 3.1.1. Globule Size and Zeta Potential

The uniformity and nanosize distribution of the globules is an important parameter to be considered during multiple nanoemulsion formulation because both these factors could enhance the solubility as well as the stability of the nanoemulsion. The droplet uniformity was indicated by polydispersity index (PDI) value. Polydispersity is actually defined as the ratio of standard deviation to average globule size. The higher the polydispersity index value is, the lower the uniformity of the globule size in the formulation [24]. A low PDI value (0.225–0.237) was obtained, which suggests the uniformity of the system [29]. The average size distribution of prepared nanoemulsions was in the range of 241.8 nm–242.5 nm (Table 2; Figure 2), which suggests the formation of nano-sized formulations [29].

Zeta potential is also an important parameter as far as the stability of nanoemulsions is concerned. In a dispersed system such as a multiple nanoemulsion, zeta potential is determined by the degree of repulsion between adjacent charged droplets, which is directly related to the system’s stability. The zeta potential value of prepared nanoemulsions was in the range of 2.88–3.61 mV. The positive value of zeta potential was accredited by the addition of chitosan [30]. The higher zeta potential value was credited to the addition of two surfactants (tween 80 and span 20) and a co-surfactant (PEG 400) to the formulation (Table 2).

#### 3.1.2. pH of the Multiple Nanoemulsions

After determining the pH of the multiple nanoemulsions by using a digital pH meter, it was found that the pH of the topical nanoemulsion was in the range of 5–6, which is considered to be the average pH of skin. Therefore, the formulations were acceptable for topical application [31] (Table 3).

#### 3.1.3. Surface Morphology

The shape of the globules must be considered in nanoemulsion formulations as it affects the performance of the formulation. Surface morphology of prepared nanoemulsions (O/W/O multiple nanoemulsion) was examined using a simple light microscope (1000×) equipped with a camera (5 megapixel). The microscopic images are shown in Figure 3. The images show the sphericity and uniformity of the globules.

#### 3.1.4. Viscosity Measurement

Viscosity is a fundamental property of a nanoemulsion that is determined by the concentrations of water, oil, and surfactants. When the water content of a nanoemulsion is increased, the viscosity of the nanoemulsion decreases. Reduced surfactant and cosurfactant concentrations, on the other hand, raise the viscosity of the nanoemulsions by increasing the interfacial tension between oil and water. Viscosity is an essential feature of nanoemulsions because it influences their stability and drug release patterns. Monitoring the stability of nanoemulsions requires assessing changes in viscosity [32]. The viscosity of chitosan-decorated simple and multiple nanoemulsions is shown in Table 4.

The measured viscosity of prepared MNE showed that this viscosity was appropriate and effective for topical drug delivery. This low viscosity provides a proper drug release as high viscosity reduces the diffusion rate of the drug from the formulation as well as the total amount of drug release and the permeation of the drug through the skin [33]. The viscosity of MNE is slightly higher than the CNE (Table 4). This is due to the double emulsion system.

### 3.2. Drug Content (%) Determination

The prepared MNE was analyzed spectrophotometrically for drug content at 262 nm. The measured % drug content of itraconazole ranged between 70 and 90%. The drug content of the prepared MNE was 90 ± 2% on average, which was higher than the SNE and CNE. This showed good drug-loading capability for the prepared MNE, which was an important requirement for the nanoemulsion [30]. The higher drug content of the MNE was due to incorporation of chitosan and the double emulsion system.

### 3.3. Percent Entrapment Efficiency Determination

The entrapment efficiency (EE) of a nanocarrier is used to estimate its efficacy in retaining the drug/active ingredient and ensuring sufficient delivery of the product to the targeted site. The method of formulation, the type of formulation ingredients and the nature of the encapsulated bioactive compound in the vesicles are all important factors that can have a significant effect on entrapment efficiency. Furthermore, as the amount of active ingredient in the nanoemulsion increases, particle size expands, lowering the nanoemulsions entrapment efficiency [34]. The entrapment efficiency of the formulated nanoemulsions were demonstrated using the centrifugation method. The obtained %EE of MNE was 45 ± 2% on average. The entrapment efficiency of ITZ increased with the MNE formulation as compared to SNE and CNE. This is due to the chitosan decoration and double emulsion system.

### 3.4. In Vitro Drug Release Study

The in vitro drug release experiment of prepared simple and multiple nanoemulsions was carried out using a Franz-type diffusion cell apparatus. In order to control premature release of the drug, chitosan-decorated multiple nanoemulsion was formulated. The result showed the controlled release of the formulation. The study was done for 24 h with an optimum level of sampling. The in vitro drug release from MNE was significantly lower than the SNE and CNE (ANOVA; *p* < 0.05). The in vitro release of simple nanoemulsion was 39.74 ± 1.8% for 24 h and that of MNE was 21.74 ± 1.2%. This might be attributed to the small droplet size of the simple nanoemulsions, which provide large surface area for drug release (Figure 4). The MNE had slightly higher viscosity and multiple layers, making it difficult for the drug to be released. Chitosan decoration also retards/controlled the release of ITZ. The incorporation of drug in the inner phase of the MNE helps in controlling release as well as protecting the drug from environmental factors. Lower release of the drug at the surface of the skin helps in higher skin retention, which is ultimately helpful in local skin infections.

### 3.5. In Vitro Drug Permeation Study

Rabbit skin as a diffusion membrane was used for the permeation study. The results showed that in vitro permeation of simple nanoemulsions was high compared to multiple nanoemulsion formulations, and this can be attributed to the low viscosity and small droplet size of the simple nanoemulsions.

The viscosity of the formulation has significant consequences for controlling the diffusion of drug through the nanoemulsion formulation into the receptor compartment. The lower the viscosity is, the higher the penetration of the drug into skin will be [28]. The results showed that SNE permeates higher through the skin compared to the CNE and MNE (Figure 5). This is attributed to the smaller droplet size and lower viscosity of SNE. Particle size is critical because it affects formulation efficacy, as a small particle size makes it a suitable carrier for drug uptake through the skin. The number of droplets that can interact on a fixed region of epidermis (*stratum corneum*) rises as particle size decreases [28]. There is good correlation between the release and permeation data for the formulated emulsions. The release of drug from the MNE was 21.74 ± 1.2%, whereas the permeation of drug across the skin was 53.33 ± 2.2%. This shows that the formulation of the drug into the MNE system helps in permeation through the skin.

### 3.6. Skin Drug Retention

The retained drug (ITZ) content in the skin was accomplished with an intention to formulate a vehicle with suitable permeation and excellent retention in the skin in order to treat topical fungal infections. Figure 6 shows that the skin drug retention of MNE is significantly higher than SNE and CNE (ANOVA; *p* < 0.05). The main features that contributed to high skin drug retention was the use of chitosan, large particle size and the high viscosity of the MNE that may result in the increased accumulation of drug in the skin. The positive charge of MNE and CNE interact with the negative charge of the skin and result in higher skin retention.

An increased skin drug retention and reduced cutaneous permeation were also investigated by [35], which developed and characterized an oil-in-water (o/w)-based hydrogel-thickened nanoemulsion using clove oil and sweet fennel for topical delivery of 8-methoxsalen (8-MOP). Low viscosity nanoemulsions are not suitable for cutaneous application. To overcome the low viscosity feature of nanoemulsions, hydrogel-thickened nanoemulsions (HTN) via chitosan (thickening agent) were formulated. Therefore, the main objectives, i.e., low skin permeation and high skin drug retention, were achieved via MNE.

### 3.7. ATR-FTIR Analysis of Skin

The effect of formulations on the skin was investigated using ATR-FTIR technique. The characteristic peaks of untreated skin appeared at 3237.13 cm^−1^, 2918.28 cm^−1^, 2850.92 cm^−1^ and 1636.55 cm^−1^. The absorption bands between 2800 and 3200 cm^−1^ matched to the alkane, aromatic hydrocarbon and amine groups. The first band was assigned to normal polymeric OH stretch. The wavenumber observed at 2918.18 and 2850.92 cm^−1^ may be assigned to C=H, while the 1636.55 cm^−1^ corresponds to the alkenyl C=C stretch [36].

In the case of epidermis, the FTIR peaks at 3287.13 cm^−1^ corresponding to an O-H and/or N-H moiety of both lipids and/or proteins of the blank or untreated skin were shifted to higher wavenumbers at 3308.07 cm^−1^, 2920.2 cm^−1^ and 3347.96 cm^−1^ and 2922.08 cm^−1^ when the skin was treated by simple and multiple nanoemulsion formulations. Similarly, the wavenumber of amide I of protein regime at 1642.62 cm^−1^ and 1643.97 cm^−1^ was also increased when treated with SNE and MNE. The combined observations suggest that SNE and MNE affectively fluidized the lipid and protein of the skin by increasing membrane permeability, which resulted in better permeation and retention of the formulation.

Similarly, the ATR-FTIR study exhibited that the dermis of skin treated with simple and multiple nanoemulsion formulations was shifted to larger wavenumbers at 3344.03 cm^−1^ and 3349.67 cm^−1^ (which correspond to O-H and/or N-H bonds of ceramide, keratin, and/or lipids). Additionally, the wavenumber of the ATR-FTIR peaks at 2920 cm^−1^ and 2852 cm^−1^ for blank dermis were increased in skin treated with SNE and MNE. It showed that MNE interacting with keratin and/or polar moieties of ceramide and lipid materials in the stratum corneum may potentially reduce the hydrogen bond strength of skin, which results in affective fluidization of the skin [37].

The results revealed that both the epidermis and dermis of the skin were affectively fluidized by the prepared nanoemulsions formulations which could possibly increase the permeation and retention of the drug. However, these effects were markedly increased by the use of two surfactants, i.e., tween 80 and PEG 400 (Figure 7)

## 4. Conclusions

Chitosan-decorated multiple nanoemulsions were prepared to incorporate ITZ for topical delivery and to control the premature release of the drug at the site of application. The formulated nanoemulsion showed good stability (no creaming and cracking), uniform size and zeta potential. The formulated MNE showed controlled release of ITZ with significantly higher skin permeation and skin drug retention as compared to SNE and CNE. ATR-FTIR analysis confirmed that MNE mainly affects the lipids and proteins of the *stratum corneum,* which results in higher permeation and retention of the drug.

It is suggested that MNE could be developed as a carrier system for topical administration of ITZ.

## Figures and Tables

**Figure 1 molecules-27-03183-f001:**
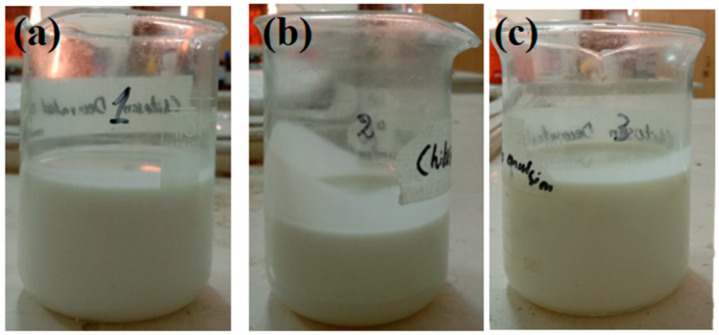
(**a**) O/W nanoemulsion, (**b**) chitosan-decorated O/W nanoemulsion and (**c**) chitosan-decorated O/W/O multiple nanoemulsion.

**Figure 2 molecules-27-03183-f002:**
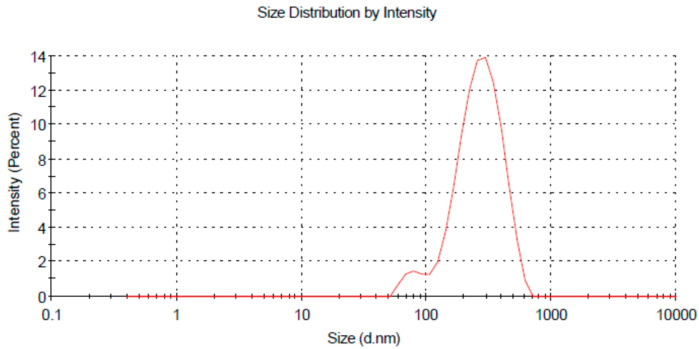
Size distribution by intensity of MNE.

**Figure 3 molecules-27-03183-f003:**
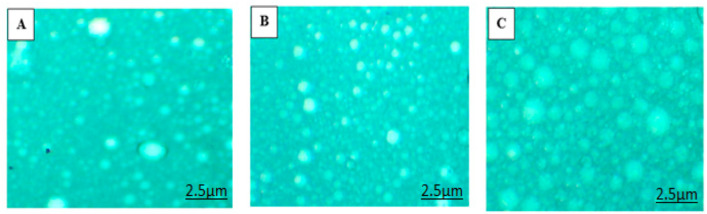
Microscopic image of formulated (**A**) simple nanoemulsions (SNE), (**B**) chitosan-decorated nanoemulsions (CNE) and (**C**) multiple nanoemulsions (MNE).

**Figure 4 molecules-27-03183-f004:**
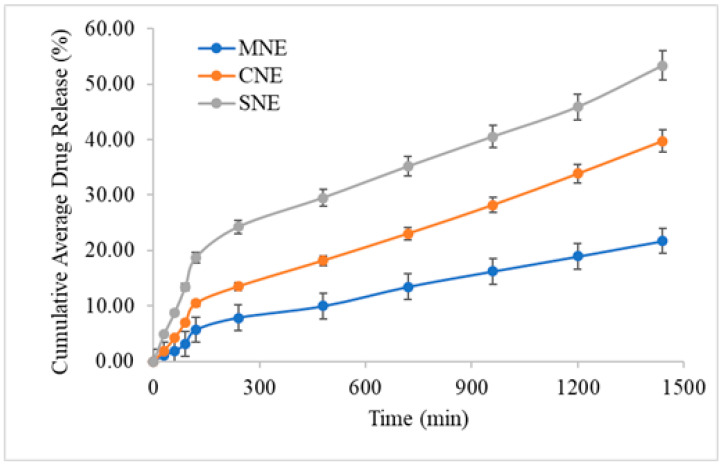
Release profiles of formulated nanoemulsions. Data were expressed as mean ± SD, *n* = 3, (ANOVA; *p* < 0.05).

**Figure 5 molecules-27-03183-f005:**
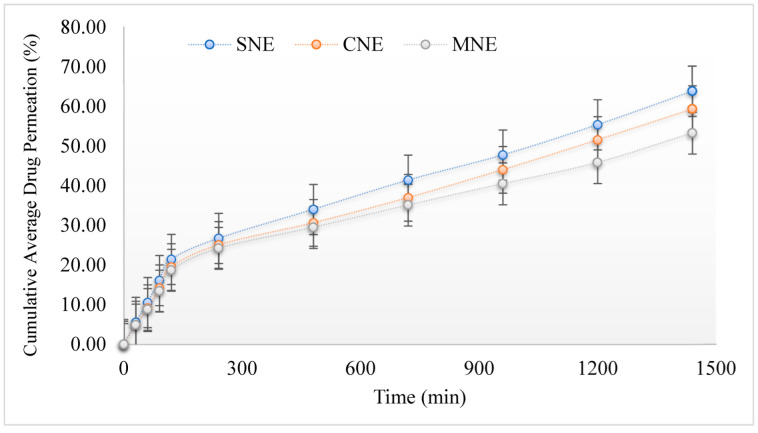
Permeation profiles of formulated nanoemulsions. Data were expressed as mean ± SD, *n* = 3.

**Figure 6 molecules-27-03183-f006:**
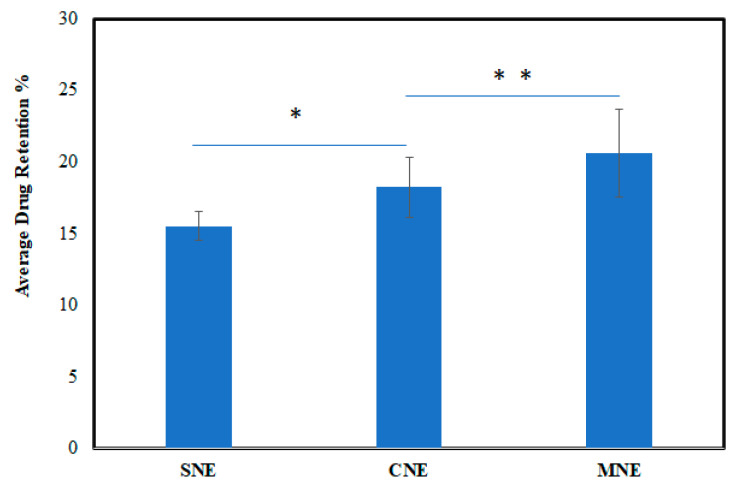
Drug retention profiles of formulated nanoemulsions. Data were expressed as mean ± SD, *n* = 3, (ANOVA; *p* < 0.05). (Note: * *p* < 0.05; ** *p* < 0.01).

**Figure 7 molecules-27-03183-f007:**
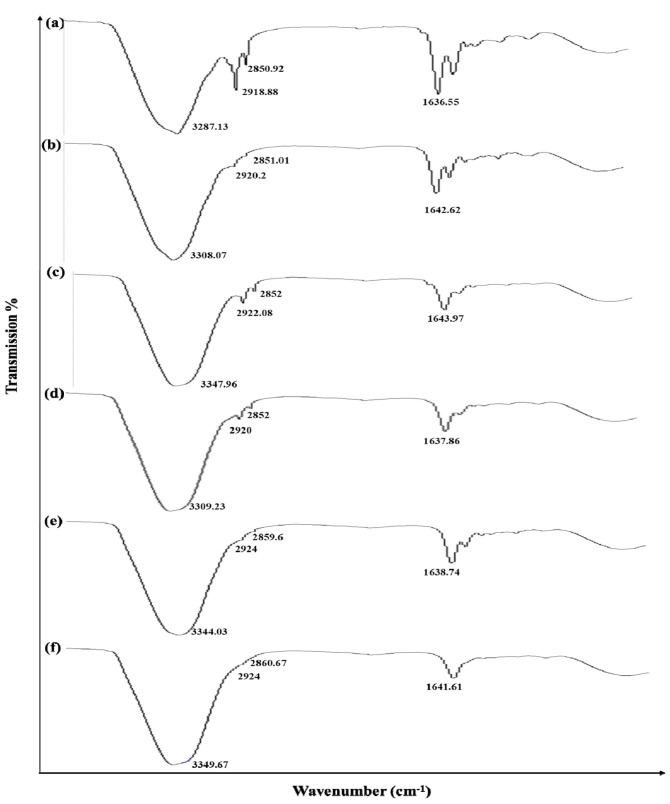
IR spectroscopy images of epidermis (**a**) untreated skin and skin treated with (**b**) SNE (**c**) MNE and dermis of (**d**) unloaded/blank skin and skin loaded with (**e**) SNE and (**f**) MNE.

**Table 1 molecules-27-03183-t001:** Composition of multiple nanoemulsions.

Phase	Components	Concentrations (%)
Inner oil phase	Olive oil	29.5
	Span 20	1
	ITZ	1
Aqueous phase	Distilled water	58.5
	PEG 400	14
	Tween 80	1
	Chitosan	250 mg
Outer oil phase	Olive oil	44
	Span 20	1

**Table 2 molecules-27-03183-t002:** Droplet size, polydispersity index (PDI) and Zeta potential of formulations.

F. Code	Droplet Size (nm)	PDI	Zeta Potential (mV)
SNE	174 ± 3.12	0.221 ± 2.21	−1.32 ± 0.41
CNE	191 ± 4.36	0.292 ± 2.38	4.32 ± 1.01
MNE	242.5 ± 5.22	0.225 ± 5.01	3.61 ± 0.79

Data were expressed as mean ± SD, *n* = 3.

**Table 3 molecules-27-03183-t003:** pH, drug content and %EE of formulations.

F. Code	pH	% Drug Content	% Entrapment Efficiency
SNE	5.5 ± 0.91	71 ± 2.02%	35.5 ± 1.52%
CNE	5.2 ± 0.45	82 ± 3.12%	41.2 ± 2.03%
MNE	5.1 ± 0.34	90 ± 2.31%	45 ± 2.09%

Data were expressed as mean ± SD, *n* = 3.

**Table 4 molecules-27-03183-t004:** Viscosity of formulated nanoemulsions.

Scheme	Chitosan-Decorated Simple o/w Nanoemulsion		Chitosan-Decorated o/w/o Multiple Nanoemulsion	
1	Spindle 1	3.31cp	Spindle 1	17.65cp
2	Spindle 2	1.44cp	Spindle 2	23.76cp

## Data Availability

The data presented in this study are available in this article.

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
