# Peer review of "Formulation and Characterization of Chitosan-Decorated Multiple Nanoemulsion for Topical Delivery In Vitro and Ex Vivo"

_molecules, 2022, doi:10.3390/molecules27103183_

Round 1

Reviewer 1 Report

The manuscript could be of interest to the scientific community, but to be published some aspects must be reviewed.

  • In general, English should be revised
  • Line 22. Please replace "...for its size..." for "... for its droplet size...".
  • Why was that homogenizationspeed and time used for primary and/or secondary homogenization? Was it optimized in some way or is it based on some previous work?
  • Although droplet sizes may be (according to some literature) included within the range of nanoemulsions, Figure 1A indicates that the system is not transparent. This condition is necessary, according to some other literature, to consider the sample as a nanoemulsion. I suggest clarifying this aspect and/or adding more definitions and references to the introduction. 
  • The apparent viscosity has been determined using a viscometer, but unless the samples are Newtonian, these values ​​are not of much interest. In addition, it should be included to which value of the shear stress or rate this viscosity value corresponds. Neither is anything known about the viscoelastic character, so the rheological characterization does not provide much information.
  • If nanoemulsions are made up of droplets... why is the term globule used?
  • Please replace "zetapotential" by "zeta potential" or "zeta-potential"
  • Figure 2. A scale is required.
  • Line 338. Without statistical analysis, I think it should be said that there is a trend.

Author Response

Dear editors,

We are extremely thankful for the kind and in-depth evaluation of our manuscript entitled “Formulation and Characterization of Chitosan Decorated Multiple Nanoemulsion for Topical Delivery: In Vitro and Ex Vivo” and pointing out the mistakes. We are highly obliged for the efforts made by reviewers, please pay our sincere appreciation to the reviewers. We thoroughly revised the whole manuscript as suggested by the reviewers. We hope that our article will be accepted for publication in this prestigious journal. The comments of the reviewers are answered in detail in this point-by-point response letter.

Reviewer 1.

  1. In general, English should be revised

Response: Do as suggested.

  1. Line 22. Please replace "...for its size..." for "... for its droplet size...".

Response: Thanks very much for the constructive comment, we have made the changes as suggested.

  1. Why was that homogenization speed and time used for primary and/or secondary homogenization? Was it optimized in some way or is it based on some previous work?

Response: The purpose of homogenization is to create a stable emulsion so that fat globules don't rise to form a cream layer. Such homogenization speed and time was used to completely disperse the oil droplets from the two emulsions into one uniform mixture. It was optimized from previous research work to form a stable nanoemulsion.

  1. Although droplet sizes may be (according to some literature) included within the range of nanoemulsions, Figure 1A indicates that the system is not transparent. This condition is necessary, according to some other literature, to consider the sample as a nanoemulsion. I suggest clarifying this aspect and/or adding more definitions and references to the introduction. 

Response: According to the research paper if the size of the nanoemulsion is < 40nm it will be transparent and if the size is >40nm it will be translucent or milky in appearance (Oliveira, C. A et al., 2018).

  1. The apparent viscosity has been determined using a viscometer, but unless the samples are Newtonian, these values ​​are not of much interest. In addition, it should be included to which value of the shear stress or rate this viscosity value corresponds. Neither is anything known about the viscoelastic character, so the rheological characterization does not provide much information.

Response: Viscosity is considered as one of the important parameters for topical dosage forms. Viscosity of the formulations was measured by viscometer. Viscosity of multiple nanoemulsion was compared with simple and chitosan decorated nanoemulsions.

  1. If nanoemulsions are made up of droplets... why is the term globule used?

Response: Do as suggested.

  1. Please replace "zetapotential" by "zeta potential" or "zeta-potential"

Response: Do as suggested.

  1. Figure 2. A scale is required.

Response: Do as suggested.

  1. Line 338. Without statistical analysis, I think it should be said that there is a trend.

Response: Thank you very much for the constructive comment, I have made the changes as suggested.

Reviewer 2 Report

The paper is interesting and the ideia is worthy of investigation. The concept of the paper is solid and the experiments were well-concepted. However, some key revisions are required before the manuscript become suitable for publication:

- English should be revised, there are a few grammar misspell

- Topical and transdermal administration are not exactly similar. There are some key differences that are not clear in the introduction section.

- ITZ abbreviation appears in the text without the meaning. The meaning of SNE, CNE and MNE appears only in the abstract. They should be on the text as well.

- How the pH was measured if it is not an aqueous solution? Was there any kind of sample preparation?

- How was possible to see the nanoparticles in a light microscope? I have a major concern here, if the nanoemulsion is really nano.

- If ITZ is not soluble in water, how did you measure %EE, in vitro release and in vitro permeation using a phospate buffer as receptor media? Also, was there any kind of sample preparation in order to destabilize the particles and to expose encapsulated ITZ?  Analytical method should be better described.

 - Discussion of results is a little poor and could be improved

- Figures 3, 4 and 5 could be improved.

- In figures 4 and 5: is there a significant difference between the groups?

- It was not clear why the use of tween 80, span 20 and PEG 400 contributes to increase skin drug retention.

Author Response

Dear editors,

We are extremely thankful for the kind and in-depth evaluation of our manuscript entitled “Formulation and Characterization of Chitosan Decorated Multiple Nanoemulsion for Topical Delivery: In Vitro and Ex Vivo” and pointing out the mistakes. We are highly obliged for the efforts made by reviewers, please pay our sincere appreciation to the reviewers. We thoroughly revised the whole manuscript as suggested by the reviewers. We hope that our article will be accepted for publication in this prestigious journal. The comments of the reviewers are answered in detail in this point-by-point response letter.

Reviewer 2.

  1. English should be revised, there are a few grammar misspell

Response: Do as suggested.

  1. Topical and transdermal administration are not exactly similar. There are some key differences that are not clear in the introduction section.

Response: yes, topical and transdermal are not similar, so I have removed the term transdermal.

  1. ITZ abbreviation appears in the text without the meaning. The meaning of SNE, CNE and MNE appears only in the abstract. They should be on the text as well.

Response: Do as suggested.

  1. - How the pH was measured if it is not an aqueous solution? Was there any kind of sample preparation?

Response: the formulated nanoemulsion was an aqueous solution, so the pH was measured using a digital pH meter and there was no sample preparation.

  1. - How was possible to see the nanoparticles in a light microscope? I have a major concern here, if the nanoemulsion is really nano.

Response: basically, the vehicle was demonstrated using the light microscope, in order to check the uniformity of globules size.

  1. - If ITZ is not soluble in water, how did you measure %EE, in vitro release and in vitro permeation using a phospate buffer as receptor media? Also, was there any kind of sample preparation in order to destabilize the particles and to expose encapsulated ITZ?  Analytical method should be better described.

Response: although ITZ is a highly lipophilic drug it is soluble in phosphate buffer pH 5.5.

  1. Discussion of results is a little poor and could be improved

Response: Discussion section was improved.

  1. Figures 3, 4 and 5 could be improved.

Response: Figures were improved.

  1. In figures 4 and 5: is there a significant difference between the groups?

Response: Figures were improved.

  1. - It was not clear why the use of tween 80, span 20 and PEG 400 contributes to increase skin drug retention.

Response: skin drug retention was increased due to the large droplet size of multiple nanoemulsions, this was not due to the use of tween 80, span 20 and PEG 400.

Round 2

Reviewer 1 Report

Taking into account the corrections carried out by the authors and the answers provided to the questions, I suggest the publication of the manuscript in Molecules.

Author Response

Dear Reviewer,

Hope you will be in good health and doing well.

Thanks for reviewing our manuscript and suggesting good comments and revisions.

Looking forward for your kind response.

Thanks and Regards

Dr. Asif Nawaz